# Lifestyle and demographic associations with 47 inflammatory and vascular stress biomarkers in 9876 blood donors

## Abstract

**Background** The emerging use of biomarkers in research and tailored care introduces a need for information about the association between biomarkers and basic demographics and lifestyle factors revealing expectable concentrations in healthy individuals while considering general demographic differences.

**Methods** A selection of 47 biomarkers, including markers of inflammation and vascular stress, were measured in plasma samples from 9876 Danish Blood Donor Study participants. Using regression models, we examined the association between biomarkers and sex, age, Body Mass Index (BMI), and smoking.

**Results** Here we show that concentrations of inflammation and vascular stress biomarkers generally increase with higher age, BMI, and smoking. Sex-specific effects are observed for multiple biomarkers.

**Conclusion** This study provides comprehensive information on concentrations of 47 plasma biomarkers in healthy individuals. The study emphasizes that knowledge about biomarker concentrations in healthy individuals is critical for improved understanding of disease pathology and for tailored care and decision support tools.

## Plain language summary

Blood-based biomarkers are circulating molecules that can help to indicate health or disease. Biomarker levels may vary depending on demographic and lifestyle factors such as age, sex, smoking status, and body mass index. Here, we examine the effects of these demographic and lifestyle factors on levels of biomarkers related to activation of the immune system and cardiovascular stress. Measurements of 47 different proteins were performed on blood samples from nearly 10,000 healthy Danish blood donors. Measurement data were linked with questionnaire data to assess effects of lifestyle. We found that immune activation and vascular stress generally increased with age, BMI, and smoking. As these measurements are from healthy blood donors they can serve as a reference for expectable effects and inflammation levels in healthy individuals. Knowledge about the healthy state is important for understanding disease progression and optimizing care.

The study of plasma biomarkers in health and disease has entered the high-throughput era, rapidly expanding into many research areas[1,2], which prompts the use of biomarkers in precision medicine. Plasma biomarkers have broad applicability as susceptibility and risk markers; as prognostic, predictive, diagnostic, and pharmacodynamic/response indicators[3]. Technological advances have led to a dramatic development in our abilities to conduct plasma biomarker studies. Thus, the simultaneous measurement of numerous analytes in a single sample and automatization make large-scale high-throughput studies feasible. Hence, recent years have seen an increasing number of large-scale biomarker studies[1,2].

Previous studies of plasma biomarkers vary greatly in number of investigated participants and measured biomarkers; and most studies have been designed to focus on specific phenotypes. Although the effect of sex, age, and body mass index (BMI) on specific biomarkers has been examined in large studies[4–6], these studies have reported on the presence of associations between biomarkers and various traits and not on absolute biomarker concentrations in specific subgroups. Further, studies with concentration levels for reference and comparison are relatively limited in sample sizes[7–9], or only include limited numbers of biomarkers (<30)[10–12]. Consequently, information on expectable concentrations is scarce for plasma concentrations of biomarkers stratified by sex, age,

✉e-mail: berkje@rm.dk

BMI, and common lifestyle factors such as smoking, i.e., phenotypes that are not exclusive to the criteria defining a healthy individual. This lack of knowledge may contribute to inequity in healthcare as plasma biomarkers, unlike genetics, are highly time-dependent and influenced by environmental and behavioral factors such as sex hormones (e.g., pre- versus post-puberty, menopause, etc.), age, BMI, and smoking.

Bridging this knowledge gap regarding the influence of common differences in demography and lifestyle factors on circulating inflammatory and vascular stress biomarkers in large cohorts represents an unmet need. A better understanding of how inflammatory and vascular stress biomarker profiles differ with age may also shed further light on the inflamm-aging hypothesis, which describes the chronic low-grade inflammation that develops with age[13,14].

In the present study, we used the Danish Blood Donor Study (DBDS) biobank and data infrastructure[15] to select samples for a large-scale, 10,000-participant, sex- and age-balanced cohort to measure inflammatory and vascular stress biomarkers. As blood donors are required to be generally healthy at the time of their donation, the DBDS represents a unique research platform and infrastructure for investigating biomarkers in healthy individuals. The primary aim of this study was to reveal associations between demographic and lifestyle factors and inflammatory and vascular stress biomarkers. The secondary aim was to provide concentrations of the individual biomarkers in healthy individuals stratified by the aforementioned factors. We find that circulating biomarker concentrations vary across strata defined by sex, age, BMI, and smoking status.

## Methods

### Participants
The DBDS is a nationwide open cohort of blood donors, aged 18–70 years at inclusion[15]. Danish blood donors must be 17–70 years, weigh more than 50 kg (60 kg for plasma donors), and be generally healthy (without diseases like cancer, severe cardiovascular disease, diabetes requiring treatment, epilepsy, and chronic viral infections like HIV, Hepatitis B and C, etc.). Additionally, several diseases and types of medicinal treatment cause temporary deferral. Participants provide informed consent, answer a questionnaire, and allow the linking of their answers with data from national registers using their unique personal identification number[16]. Currently, more than 161,000 donors are included, and more than 400,000 questionnaires, distributed across several versions, have been answered[17]. The questionnaires have addressed behavioral and phenotypic features such as BMI, smoking status, stress, migraine, alcohol consumption, menopause, the 12-item Short-Form Health Survey (SF-12), and allergy. Detailed information on the DBDS cohort is available in previous publications[15,18]. For every participant, a gel-separated plasma sample is stored at study inclusion and at every blood or plasma donation. Thus, the DBDS encompasses a biobank with more than two million blood samples.

### Selection of participants for the present study
Participants for the *DBDS 10 K Inflammatory Biomarker Cohort* were selected among all DBDS participants who had inclusion samples stored in the biobank. The first 982 samples were part of a pilot study before the final selection procedure was decided and, thus, introduced a small bias among samples stored before 2017. The pilot study was followed by an additional +9000 samples. A random inclusion sample was chosen for each participant at the time, yielding 128,017 available samples. Among the available samples, 10,277 were picked, ensuring equal numbers of participants in sex- and age-stratified groups. The age groups used were: 18–29, 30–39, 40–49, 50–59, and 60–69 years. We excluded 1383 samples that could not be identified, for which concerns were raised regarding sample quality (freeze-thaw cycles, low sample volume, caps that had become loose in storage), or that were from donors who withdrew their consent before the final data analysis. A total of 9876 samples were included in the final cohort. The selection is summarized in Fig. 1.

### Sample handling
Samples were collected in gel-separated ethylene diamine tetra-acetic acid (EDTA) tubes before blood donation, centrifuged, frozen on the same day, and stored in the primary tube at −22 °C. Samples selected for analysis were thawed and aliquoted to 96-well plates on an automated platform (Hamilton MicroLab STAR liquid-handling platform, Reno, USA) at the Department of Clinical Immunology, Copenhagen University Hospital, Rigshospitalet, Copenhagen, Denmark. 96-well plates with the plasma samples were refrozen until the time of analysis.

### Biomarker measurements
Biomarker measurements were performed using the electrochemiluminescence-based Meso Scale Discovery (MSD) V-PLEX Human Biomarker 54-plex kit. The assay comprises seven individual multiplex panels (Table 1). Panels were analyzed according to the manufacturer's instructions. Several of the TH17 panel 1 biomarkers had extensive quality control issues that could not be resolved despite numerous efforts from MSD and the involved laboratories. The TH17 panel measurements were therefore discontinued and excluded from further analysis. Panels were measured at the Department of Clinical Immunology, Aarhus University Hospital, Aarhus, Denmark; the Department of Clinical Immunology, Copenhagen University Hospital, Rigshospitalet, Copenhagen, Denmark, and at the Department of Health Technology, the Technical University of Denmark, Lyngby, Denmark (Table 1).

All panels were analyzed on the Meso QuickPlex SQ 120 platform (Meso Scale Diagnostics LLC, Maryland, USA). Duplicates of internal control (pooled plasma spiked with calibrators from the seven different panels and stored at −80 °C until use) were analyzed, and a standard curve was drawn for each plate. A negative control was prepared using only the diluent. Measurements were performed from October 2020 to February 2022. The MSD kit LOT numbers used for the study are listed in Supplementary Table 1.

Quality control was performed in two steps. Initial QC was performed in the MSD Discovery Workbench post-analysis ensuring sufficient quality of the standard curve used for concentration estimation. Points on the standard curve were excluded if the recovery mean was outside a range of 80–120%, or if CV for a point was higher than 20%. If more than three points on the standard curve were excluded the entire curve was substituted for a curve from a plate analyzed on the same day. This step was advised by the manufacturer and is described in the protocol.

The second part of the quality control excluded biomarker measurements if the internal control was not within ±3 SD from the mean of internal controls for all plates in the study (Supplementary Fig. 1). This excluded a total of 469 measurements. Measurements categorized as "Below Fit Curve

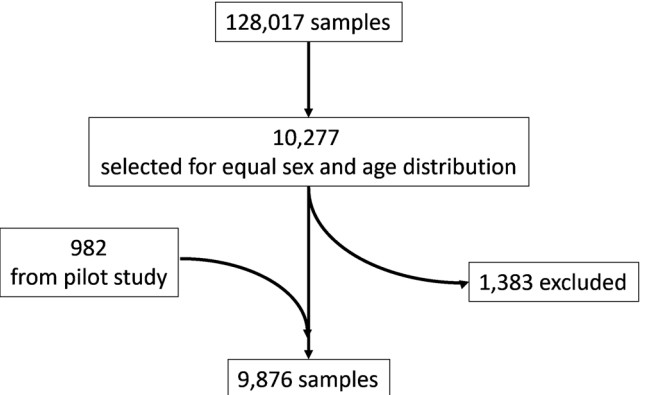

**Fig. 1 | Flow chart of the sample selection.** A random inclusion sample was selected for every participant and the selection was sampled to ensure a subset with equal age and sex distribution. Samples from the pilot study were added to the pool, while samples that could not be identified, were from participants who withdrew consent, or had concerns regarding sample quality were excluded.

Range" by the MSD software, reported as NA, were assigned a concentration by randomly drawing from a uniform distribution between 0 and the lower detection limit. Measurements categorized as "Above Fit Curve Range" were set to the upper detection limit. To account for plate-to-plate variation, we did median normalization (**Formula 1**).

**Formula 1:**

$$concentration_{normalized} = \frac{concentration}{median_{plate}} \cdot median_{overall}$$

IL-8 and VEGF-A was measured on two panels. Here we used IL-8 from the proinflammatory panel 1 and VEGF-A from Angiogenesis Panel 1, as these had the most measurements available within the detection range (Supplementary Data 1).

Technical covariates such as sample age, regional differences, storage times, and storage temperatures were investigated to clarify possible bias. Samples were collected all over Denmark, with slight differences in the protocols. Supplementary Fig. 2 shows regional differences.

Samples were collected between the year 2010 and 2021. Prolonged storage can affect the quality of the sample and, thereby also the measurements. Supplementary Fig. 3 illustrates the effect of storage time, while Supplementary Data 2 displays the estimated effect of sample age adjusted for region and analysis date. Samples were stored at −20 C° for four out of five regions, while the last region (South Denmark (S)) stored samples at −80 C°. There were significant regional differences for some markers, as displayed in Supplementary Data 4, and consequently, the region was included as an adjustment.

The literature has shown differences in the concentrations of several inflammatory biomarkers depending on the circadian rhythm[19], and Supplementary Fig. 4 shows a difference in mean time of day between age groups. We observed that older donors (≥60 years) donated earlier in the day, while the youngest donors (<30 years) donated later in the day (mean 12:09 vs 12:50 PM, Wilcoxon rank-sum test $p < 0.001$, Supplementary Fig. 4). An extensive investigation of this is beyond the scope of this article, but adjusted linear models showed some notable differences for biomarkers such as increased CCL26, CRP, FLT-1, IL-6, and IL-12 over the day. In contrast, the following biomarkers decreased over the day: CCL2, CCL3, CCL4, CCL13, CXCL10, IFNγ, IL-5, IL-7, IL-8, IL-10, IL-15, PlGF, sVCAM-1, TNFα, and TNFβ. Supplementary Fig. 5 shows how each biomarker changed during the day in an unadjusted fit created by smoothed conditional means and local polynomial regression fitting.

Density plots of the log of median transformed concentrations for each assay are displayed in Supplementary Fig. 6.

Calculated concentrations are reported in pg/mL except for C-reactive protein (CRP) and Serum amyloid A (SAA), which are reported in mg/L, and sVCAM-1 and sICAM-1, which are reported in ng/mL.

To improve the biological interpretation of our results, the biomarkers were grouped by function or associated cell types of origin, as displayed in Table 2.

## Statistics

**Analysis model.** Linear regression models were used to assess the effects of the exposures (sex, age, BMI, and smoking) on biomarker concentrations, with relevant adjustments among case-complete participants. The following covariates were included in the models: sample age (time stored in freezer from donation to analysis), sex, age (as a continuous variable), BMI (as a continuous variable), smoking, administrative region in Denmark, and measurement date. The causal structure of the models is illustrated as directed acyclic graphs in Supplementary Fig. 7. To assess sex-specific effects, models were constructed, including an interaction term between exposure and sex. We considered the direct effect of sex to be of primary interest and therefore estimated sex effects adjusted for the mediators BMI and smoking. Visual model diagnostics were performed by plotting residuals and inspecting them for all main models.

**Biomarker and data presentation.** Results are presented as medians with interquartile ranges (IQR) or as numbers and proportions, as appropriate. Tests were two-sided. The biomarker concentrations were log-transformed, and estimates were transformed back to a proportion for the data presentation as percentage change with 95% confidence intervals (CI).

For the heatmaps, the concentrations were scaled and median concentrations for each category (e.g., age group or BMI group) per biomarker

**Table 1 | The Meso Scale Discovery (MSD) panels comprising the MSD V-PLEX Human Biomarker 54-plex kit and their biomarkers**

| MSD panel (all human) | Measured at | Dilution factor | Biomarkers |
|---|---|---|---|
| Proinflammatory Panel 1 | RH | 1:2 | IFN-γ, IL-1β, IL-2, IL-4, IL-6, CXCL8/IL-8, IL-10, IL-12p70, IL-13, TNF-α |
| Cytokine Panel 1 | RH | 1:2 | GM-CSF, IL-1α, IL-5, IL-7, IL-12/IL-23p40, IL-15, IL-16, IL-17A, TNF-β, VEGF-A |
| Cytokine Panel 2 | DTU | 1:4 | IL-17A/F, IL-17B, IL-17C, IL-17D, IL-1RA, IL-3, IL-9, TSLP |
| TH17 Panel 1 (excluded) | AUH | 1:4 | IL-17A (Gen.B) IL-21, IL-31, IL-27, IL-23, IL-22, MIP-3α |
| Angiogenesis Panel 1 | AUH | 1:2 | FGF (basic), VEGFR-1/Flt-1, PIGF, Tie-2, VEGF-A, VEGF-C, VEGF-D |
| Chemokine Panel 1 | DTU | 1:4 | CCL11 (Eotaxin), CCL4 (MIP-1β), CCL26 (Eotaxin-3), CCL17 (TARC), CXCL10 (IP-10), CCL3 (MIP-1α), CXCL8/IL-8, CCL2 (MCP-1), CCL22 (MDC), CCL13 (MCP-4) |
| Vascular Injury Panel 2 | DTU | 1:1000 | SAA, CRP, sVCAM-1, sICAM-1 |

*RH* Rigshospitalet, University of Copenhagen, *AUH* Aarhus University Hospital, University of Aarhus, *DTU* the Technical University of Denmark.

**Table 2 | Biomarker grouping by function and associated cell types of origin**

| Biomarker groups | Individual biomarkers |
|---|---|
| Proinflammatory | CRP, SAA, TNF-α, TNF-β, IL-1α, IL-1β, IL-1RA, IL-3, IL-6, IL-12/IL-23p40, IL-12p70, IL-15 |
| T cell derived | IFN-γ, IL-2, IL-4, IL-5, IL-9, IL-10, IL-13, IL-16, IL-17A, IL-17A/F, IL-17B, IL-17C, IL-17D |
| Chemokines | CCL11, CCL26, CXCL8/IL-8, CXCL10, CCL2, CCL13, CCL22, CCL3, CCL4, CCL17 |
| Growth factors and vascular | bFGF, Flt-1, Tie-2, IL-7, PIGF, VEGF-A, VEGF-C, VEGF-D, TSLP, GM-CSF, sICAM-1, sVCAM-1 |

For an easier data presentation, we grouped the various biomarkers as shown.

were calculated. We then used the Ward.D2 algorithm for hierarchical clustering of the rows, of which a distance matrix consisting of Euclidean distance measures was used. Significance levels in the heatmaps were pulled from the results of linear models displayed in Supplementary Tables 3–49. Radar charts were generated and plotted with scaled values (0–1) for the log of the concentration values for each marker stratified by sex. Scales on radar charts were set to the first quartile (0% on the charts) and third quartile (100% on the charts) of the scaled values within each biomarker and sex. The points in the radar charts thus represent the median of the scaled value for the examined group for each given marker, e.g., the median of the scaled value in smokers is shown beside the median for non-smokers. Differences displayed on the radar charts were compared by the linear models described above. Tables with concentration median for age groups, BMI, and smoking were displayed with IQR and with a percentage of the mean in each group compared with the reference.

**Covariates**. BMI was calculated from self-reported height and weight. BMI was investigated with participants grouped by WHO categories (<18.5 kg/m$^2$: underweight, ≥18.5 kg/m$^2$ and <25 kg/m$^2$: normal weight, ≥25 kg/m$^2$ and <30 kg/m$^2$: overweight, ≥30 kg/m$^2$: obese). Current smoking behavior was defined as those who reported smoking at least once a week, excluding e-cigarette users.

Analysis was performed in R version 4 (R Foundation for Statistical Computing, Vienna, Austria, www.R-project.org). $P$ value significance level was Bonferroni adjusted by the number of biomarkers in the relevant biomarker group to 0.00416 for proinflammatory and growth factor biomarkers, 0.00385 for T cell-derived biomarkers, and 0.005 for chemokines. Visual model diagnostics were performed by plotting residuals and inspecting leveraging for all main models.

### Ethics
Oral and written informed consent was obtained from all study participants. The DBDS was approved by the Danish Data Protection Agency (P-2019-99) and the Committees on Health Research Ethics in the Central Denmark Region (1-10-72-95-13) and the Zealand Region (SJ-740). The ethical approvals covered the measurement of biomarkers in the DBDS, and additional ethical approval was therefore not required.

### Reporting summary
Further information on research design is available in the Nature Portfolio Reporting Summary linked to this article.

## Results
### Cohort demographics
Demographics of the 9876 cohort participants are described in Table 3. In alignment with the sample selection algorithm, similar numbers were achieved in the age- and sex-stratified groups. BMI and smoking reflected previous findings in Danish blood donors[20]. The age distribution of smokers is shown in Supplementary Table 2. The distribution of participants across Danish administrative regions reflected inclusions in the DBDS cohort. The median number of prior donations at the date of the sample, including plasma donations, was 17 (IQR 7–32). In total, 602 participants had missing BMI or smoking information. Only few participants were underweight (59 participants, 0.006%), and these were therefore excluded from the BMI analysis. The number of measurements from participants with questionnaire information available for adjusted analysis is shown in Supplementary Data 1.

### Biomarker measurements
In the present study we aimed to provide reliable and comparable concentrations for a wide range of inflammatory and vascular stress biomarkers across sex, age, smoking, and BMI. The heatmaps and the radar charts (Figs. 2–7) provide a quick impression of the biomarker profile differences according to the examined traits, whereas those with a need for more detailed information may consult the more detailed Supplementary

Tables 3–49 with further information on individual biomarkers and their associations with demographics and lifestyle factors. For mean and standard deviation in each age group, we refer to Supplementary Data 5.

### Available measurements
Only few measurements were above the upper limit of detection (16 SAA measurements and 12 measurements distributed across bFGF, IL-16, VEGF-A, CCL26, CXCL10, CCL17, and IL-1RA). For the biomarkers IL-3, IL-1B, IL-13, IL-12p70, IL-4, IL-2, IL-9, IL-17A/F, IL-17C, TSLP, and GM-CSF (ordered by % below detection range), more than 10% were below detection range (see supplementary Data 1).

### Prior donations, regional differences, and sample age
There were 13 biomarkers significantly associated with sample age (Supplementary Data 2), and sample age was consequently included as an adjustment in subsequent models. We examined the effect of prior donations (all types) within three years and only found an effect for IL-15 and IL-7 and did not pursue this further (Supplementary Data 3). Concentrations of several markers differed between regions (Supplementary Data 4), and thus region was included as an adjustment in all other models.

### Association between sex and age and biomarker concentrations
**Proinflammatory biomarkers**. Most proinflammatory biomarker concentrations differed between sexes (CRP, SAA, IL-6, and IL-12/IL-23p40 concentrations were higher in females; TNF-α and TNF-β levels were higher in males), with IL-6 and IL-15 having a greater increase with age in males, as presented for the interaction term in Fig. 2 and in Supplementary Tables 3–14 found at the end of the supplementary information containing various regression results for each biomarker. Several proinflammatory biomarkers increased with age (SAA, IL-6, and IL-15) and were thus higher in older participants of both sexes (Figs. 2, 5a). TNF-β and IL-12/IL-23P40 decreased with age in both sexes. CRP increased with age in males and TNF-α increased with age in females. CRP was highest in young females, and lowest in 40–49-year-old females, whereafter concentrations increased with age (Supplementary Fig. 8).

**T cell-derived biomarkers**. Among T cell-derived biomarkers, IL-16 differed between sexes, and sex-specific age effects for IL-5 and IL-17B (Fig. 2 and Supplementary Tables 18, 22, 25) was observed. IFN-γ, IL-5, IL-9, and IL-17D increased with age in both sexes, with a greater IL-5 increase in males. IL-17A decreased with age in both sexes. IL-17A/F and IL-17B increased with age in females. IL-10 and IL-16 decreased with age in males.

**Chemokine biomarkers**. Most chemokines differed between sexes (CCL2, 4, 11, 13, 17, and 26, CXCL10, and IL-8 were higher in male participants, whereas CCL22 was higher in females), with a greater increase with age in females than in men for CCL11, CCL2, CCL13, and CXCL10 (Fig. 2 and Supplementary Tables 28–37). All chemokine concentrations increased with age, except CCL26, which remained unchanged (Figs. 2, 5c).

**Growth factors and vascular biomarkers**. Several growth factors and vascular biomarkers differed between sexes (Flt-1, PlGF, sICAM-1, sVCAM-1, and Tie-2 higher in males. IL-7 and VEGF-D higher in females) with a greater Flt-1 and PlGF increase with age in females and a VEGF-D increase in males (Fig. 2 and Supplementary Tables 38–49). Furthermore, bFGF, PlGF, TSLP, VEGF-A, VEGF-C, and VEGF-D increased with age in both sexes. Flt-1, Tie-2, sICAM-1, and sVCAM-1 increased with age in females, whereas Tie-2 and sVCAM-1 decreased with age in males. IL-7 decreased with age in females (Figs. 2, 5d).

In Fig. 2, clusters of biomarkers appeared to display distinct patterns across sex and age, with the proinflammatory, T cell-derived, and growth factor and vascular biomarkers generally clustering away from the

**Table 3 | Demographics of the cohort**

| | Male N = 4933 | | Female N = 4943 | | Total N = 9876 | | Missing |
|---|---|---|---|---|---|---|---|
| | **N** | **%** | **N** | **%** | **N** | **%** | |
| **Age** | | | | | | | |
| Years, median (IQR) | 45 (32–57) | | 45 (32–57) | | 45 (32–57) | | 0 |
| 18–29 | 991 | 20.1 | 1003 | 20.3 | 1994 | 20.2 | |
| 30–39 | 1010 | 20.5 | 1000 | 20.2 | 2010 | 20.4 | |
| 40–49 | 1011 | 20.5 | 1001 | 20.2 | 2012 | 20.4 | |
| 50–59 | 994 | 20.2 | 992 | 20.1 | 1986 | 20.1 | |
| 60+ | 927 | 18.8 | 947 | 19.2 | 1873 | 19.0 | |
| **Smoking** | | | | | | | |
| Current smokers | 618 | 13.2 | 653 | 13.9 | 1,271 | 13.6 | 513 |
| **Anthropometric Measurements** | | | | | | | |
| Height (cm), median (IQR) | 182 (178–186) | | 168 (165–172) | | 175 (168–182) | | 475 |
| Weight (kg), median (IQR) | 85 (77–93) | | 69 (63–78) | | 78 (68–88) | | 504 |
| BMI, median (IQR) | 25.4 (23.6–27.8) | | 24.3 (22.2–27.3) | | 24.9 (22.8–27.7) | | 532 |
| **Donation History & Sample Data** | | | | | | | |
| Donations in last three years, median (IQR) | 5 (3–8) | | 4 (2–7) | | 5 (3–7) | | 0 |
| Sample age (years), median (IQR) | 7.9 (3.6–9.6) | | 7.6 (3.5–9.5) | | 7.7 (3.5–9.6) | | 0 |
| **No. of participants per region** | | | | | | | |
| Capital | 1976 | 40.1 | 2016 | 40.8 | 3992 | 40.4 | 0 |
| Central Denmark | 1315 | 26.7 | 1294 | 26.2 | 2609 | 26.4 | |
| North Denmark | 541 | 11 | 503 | 10.2 | 1044 | 10.6 | |
| Southern Denmark | 357 | 7.2 | 301 | 6.1 | 658 | 6.7 | |
| Zealand | 744 | 15.1 | 829 | 16.8 | 1573 | 15.9 | |

Numbers are presented as count with % of total or as median with IQR.

chemokine cluster. As an exception, IL-8 clustered with proinflammatory and T cell-derived biomarkers. To examine the possible effects of menopause, we compared females aged 50 or more to those aged below 50 and found most chemokines to be increased after age 50 while only some proinflammatory, T cell-derived, and growth factors and vascular biomarkers showed differences (Supplementary Data 6).

**Association between BMI and biomarkers**
**Proinflammatory biomarkers.** Comparing participants with a normal BMI with obese participants, CRP, IL-6, and TNF-α displayed a greater increase in females, as shown by the interaction terms in Fig. 3 and in Supplementary Tables 3, 8, and 13. Compared with the normal BMI group, CRP, and IL-6 concentrations increased with BMI group in both sexes; SAA, IL-1RA, IL-12/IL-23p40, TNF-α, and TNF-β increased with BMI group in male participants (Figs. 3, 6a and Supplementary Tables 3–14). TNF-α was increased and IL-15 was decreased in obese females.

**T cell-derived biomarkers.** IL-17A tended to increase with BMI group in males and IL-17C was increased in overweight males (Figs. 3, 6b and Supplementary Tables 23, 26).

**Chemokine biomarkers.** The effect of obesity on concentrations of CCL11 and CCL4 was greater in males (see interaction terms in Fig. 3 and Supplementary Tables 30, 31). CCL2, CCL3, CCL22, and CXCL10 increased with BMI group in males; IL-8, CCL4, CCL13, and CCL26 were higher in obese males only; CCL3 was increased in obese females (Figs. 3, 6c and the Supplementary Tables 28–37).

**Growth factors and vascular biomarkers.** Although no association was found with BMI for VEGF-D, the effect of BMI on concentrations of VEGF-D differed between males and females, as displayed by the interaction terms in Supplementary Table 49. PlGF, sICAM-1, and VEGF-A increased with BMI group in males, and bFGF, IL-7, TLSP, and VEGF-C were increased in obese males (Figs. 3, 6d and Supplementary Tables 38–49).

In Fig. 3, it appears that clusters of biomarkers displayed distinct patterns across sex, age, and BMI group, with several proinflammatory and T cell biomarkers clustering together and the chemokine cluster, the growth factor, and the vascular biomarker cluster being distinct. Analysis of BMI as a continuous variable stratified by age group and sex generally reflected the analysis of BMI groups, although additional significant associations were observed in females (supplementary Data 7) compared to analysis using BMI groups (Supplementary Tables S3–S49). This was particularly true for assays in the chemokine and growth factor groups.

**Association between smoking and biomarkers**
**Proinflammatory biomarkers.** Compared with non-smokers, smokers' concentrations of IL-1β, IL-1RA, IL-6, and IL-15 were higher in both sexes (Figs. 4, 7a and Supplementary Tables 3–14). CRP and IL-1α were higher in male smokers. IL-12/IL-23p40 and TNF-β concentrations were lower in both male and female smokers, and SAA was lower in female smokers.

**T cell-derived biomarkers.** Compared with non-smokers, smokers' IL-2 and IL-17A concentrations were lower in both sexes (Figs. 4, 7b and Supplementary Tables 15–27). IL-16 concentrations increased with smoking in both sexes, whereas IL-5 and IL-17D increased with smoking in males and IL-17C increased in females. IFN-γ was lower in male smokers. IL-17A/F and IL-17B was lower in female smokers and with a sex-specific effect for IL-17B.

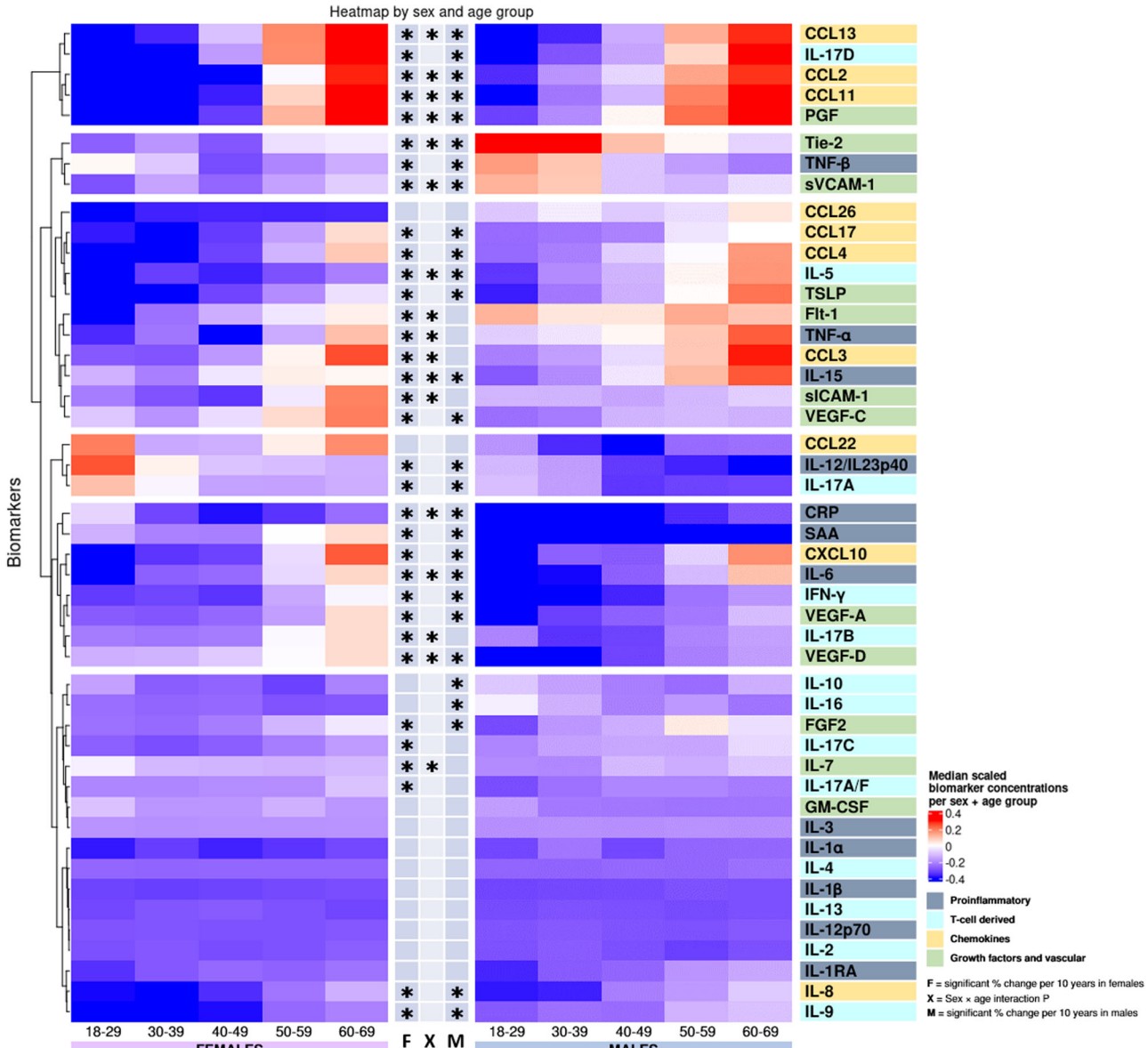

**Fig. 2 | Heatmap of median scaled biomarker concentrations by sex and age group.** Biomarker labels are color-coded based on their grouping. Biomarkers are clustered by change with age. *Indicates significant change with age in females (F), males (M), or significant interaction between age and sex (X) derived from the linear models in the Supplementary Tables 3–49 using log-transformed concentrations and adjusted for smoking, BMI, region, sample storage time and measurement date, as relevant.

**Chemokine biomarkers.** Compared with non-smokers, smokers' concentrations of CCL11, IL-8, CCL2, CCL22, and CCL17 were higher in both sexes, whereas CXCL10 was lower in smokers of both sexes (Figs. 4, 7c and Supplementary Tables 28–37). CCL26 was higher in male smokers, whereas CCL3 and CCL13 was higher in female smokers.

**Growth factors and vascular biomarkers.** Compared with non-smokers, smokers' PlGF, sICAM-1, TLSP, VEGF-A, and VEGF-D concentrations were higher, whereas sVCAM-1 and Tie-2 tended to decrease with smoking in both sexes (Figs. 4, 7d and Supplementary Tables 38–49). VEGF-C was higher in male smokers.

In Fig. 4, clusters of biomarkers appear to be displayed in distinct patterns across sex, age, and smoking, with the proinflammatory and T cell-derived clustering together and chemokines and growth factor and vascular biomarkers clustering together. After adjusting for multiple comparisons, we found no statistically significant interaction between sex and smoking, despite observing that certain biomarkers (e.g., CCL13, CCL26, IL-1α, IL-5,

IL-12p70, IL-17B, IFNγ, SAA, and CRP) were significantly correlated with smoking in only one sex.

## Discussion

Many of the investigated biomarkers increased with age across the various biomarker groupings. Previous studies are conflicting as they have reported angiogenesis and VEGF concentrations as either reduced[21] or increased[6,22] with age.

The finding that most biomarkers of inflammation and vascular stress increased with age is in line with the inflamm-aging hypothesis[13,14]. We also found several biomarkers to decline with age, including IL-10, IL-12/IL-23P40, IL-16, IL-17A, and TNF-β. Importantly, the anti-inflammatory biomarker IL-10 decreased with higher age. In contrast to our findings, IL-17 and IL-12/IL-23P40 have previously been reported to increase with age in healthy individuals though the oldest age group in these studies was older than our oldest group. Because our cohort is likely healthier with less comorbidity, the cohorts may not be completely comparable[23,24]. Selecting

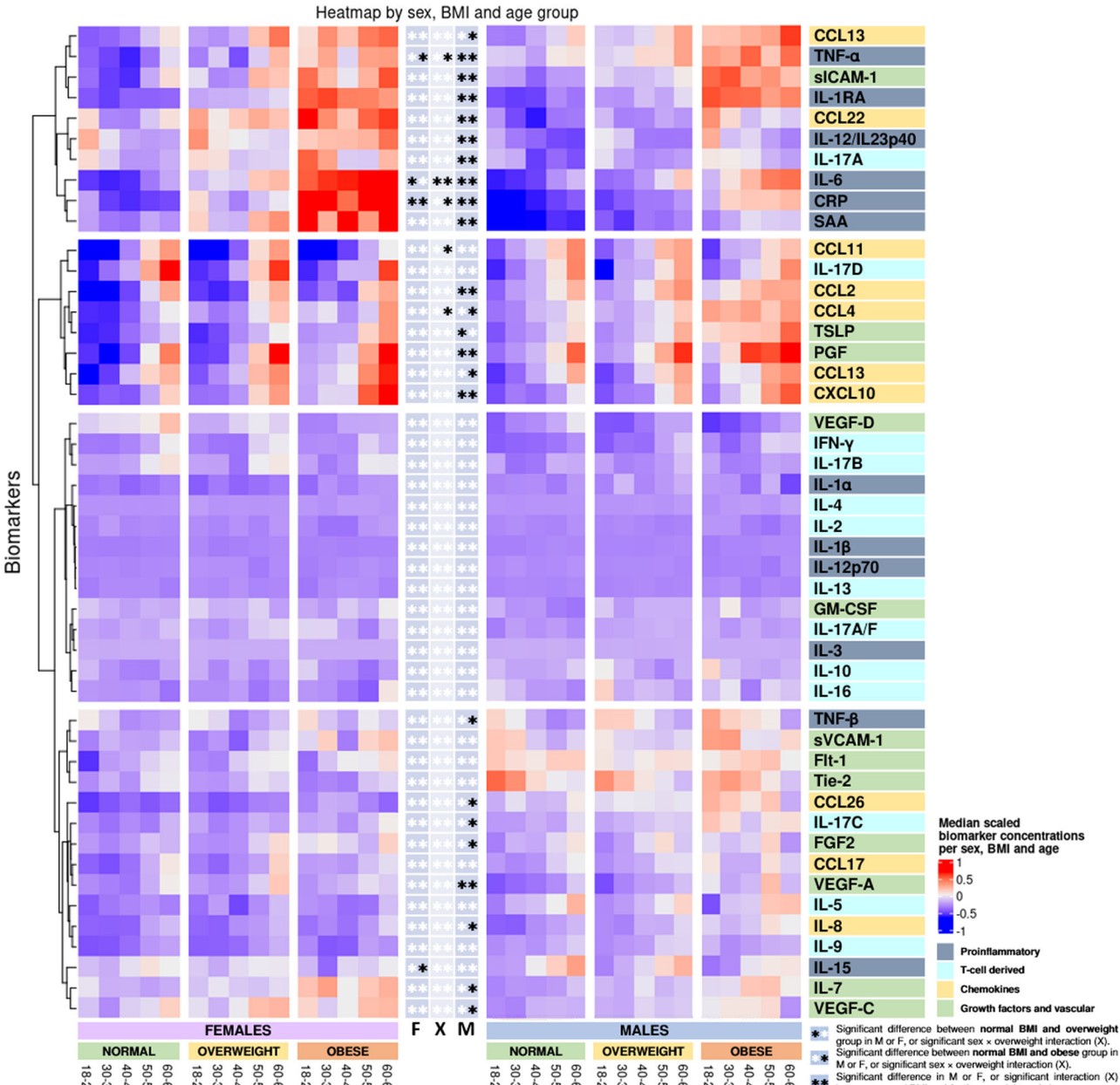

**Fig. 3 | Heatmap of differences in biomarker concentrations by sex and BMI group.** Biomarkers are clustered by change with BMI group. *Indicates significant change in biomarker concentrations between normal BMI group and either the overweight group (star to the left) or the obese group (star to the right) in females (F), males (M), or significant interaction between BMI group and sex (X) derived from the linear models in the Supplementary Tables 3–49. **Indicates a significant change in biomarker concentrations between the normal BMI group and the overweight group, and additionally between the normal group and the obese group. Linear regressions were performed using log-transformed concentrations and adjusted for sex, age, smoking, region, sample storage time, and measurement date.

samples with an equal age distribution may introduce a slight survivor bias in the oldest groups.

CRP was higher in young females, lower in middle-aged females, and higher in older females (50–60 years and 60–70 years). This is likely explained by the use of oral contraceptives, mainly among young women, as reported by us previously[25]. Biomarker changes relating to menopause have been reported[26]. Questions on menopause and contraceptives were only included in the first questionnaire round and were thus only available for a subset of participants in this study. Our observations support the notion that some of the changes in females with age appear around menopause.

For many of the investigated biomarkers, large proteomics studies have also reported on the influence of sex and age. These studies did not, however, report the actual concentrations of the biomarkers but the associations they

found generally corroborate our findings[6,27]. The wide age span, even sex distribution, and healthy state of blood donors facilitate the usage of the reported concentrations of inflammatory and vascular stress biomarkers in future studies.

We found that higher BMI strongly influenced many of the biomarkers investigated. Markedly a higher BMI was associated with higher concentrations of proinflammatory biomarkers and chemokines. Also, several growth factors and vascular biomarkers increased with BMI, potentially reflecting that an increased BMI is associated with enhanced angiogenesis with the increase in body mass. The relationship between inflammation, angiogenesis, and overweight/obesity is well known[28]. In line with our findings, other studies have reported higher concentrations of several proinflammatory biomarkers in overweight and obese individuals[29–33]; this is in line with the notion that adipose tissue is a high producer of

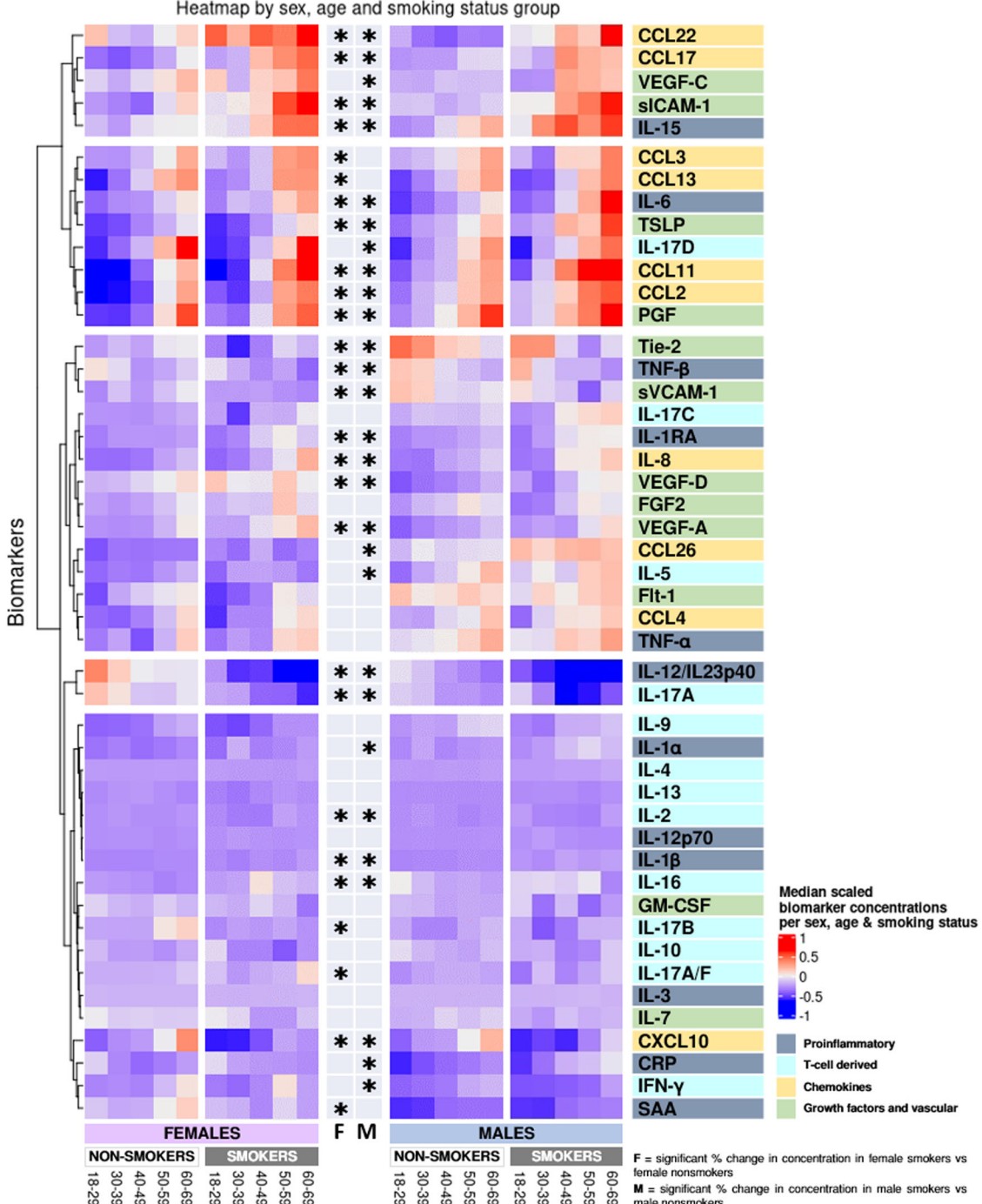

**Fig. 4 | Heatmap of differences in biomarker concentrations per sex and smoking status.** Biomarkers are clustered by change with smoking. *Indicate significant change with smoking in females (F), males (M) derived from the linear models in Supplementary Tables 3–49 where log-transformed concentrations were used, adjusted for age, BMI, region, sample storage time, and measurement date.

proinflammatory molecules, hereby driving systemic inflammation[34,35]. Notably, most associations between higher BMI and biomarkers were found in males, which may be explained by sex-specific (sex-hormonal) differences in response to increased adipose tissue. This difference in how the sexes are affected by increased BMI is known and possibly conveyed by general differences in the deposition of adipose tissue[36,37]. When BMI was examined as a continuous variable, we found additional associations among females, but the results generally reflected the analysis presented in Supplementary Tables 3–49.

The age distribution of smokers in the DBDS resembled that of previous reports for the Danish population[38]. Several studies have reported

changes in inflammatory and vascular stress biomarkers related to tobacco smoking, with most studies reporting that smoking increases concentrations of proinflammatory, angiogenesis, vascular, and oxidative stress biomarkers, but decreases the concentration of anti-inflammatory biomarkers such as IL-10[39–43].

We found smoking to be associated with higher concentrations of proinflammatory, chemokine, growth factor, and vascular stress biomarkers and with lower concentrations of T cell-derived biomarkers. These findings support the well-known proinflammatory effect of smoking and emphasize that smoking induces T cell suppression or dysfunction, possibly related to nicotine that has known immunosuppressive effects[44,45]. Previous smoking

**Fig. 5 | Radar charts of the median of the scaled concentration of biomarkers for the age groups 18–30 and 60–70 years.** Axis percentage spans from the first to the third quartile within each marker. **a** is proinflammatory group, **b** is T cell-derived group, **c** is chemokine group, and **d** is growth factors and vascular group. * marks confidence intervals not spanning 0 for the effect of a 10-year increase in age.

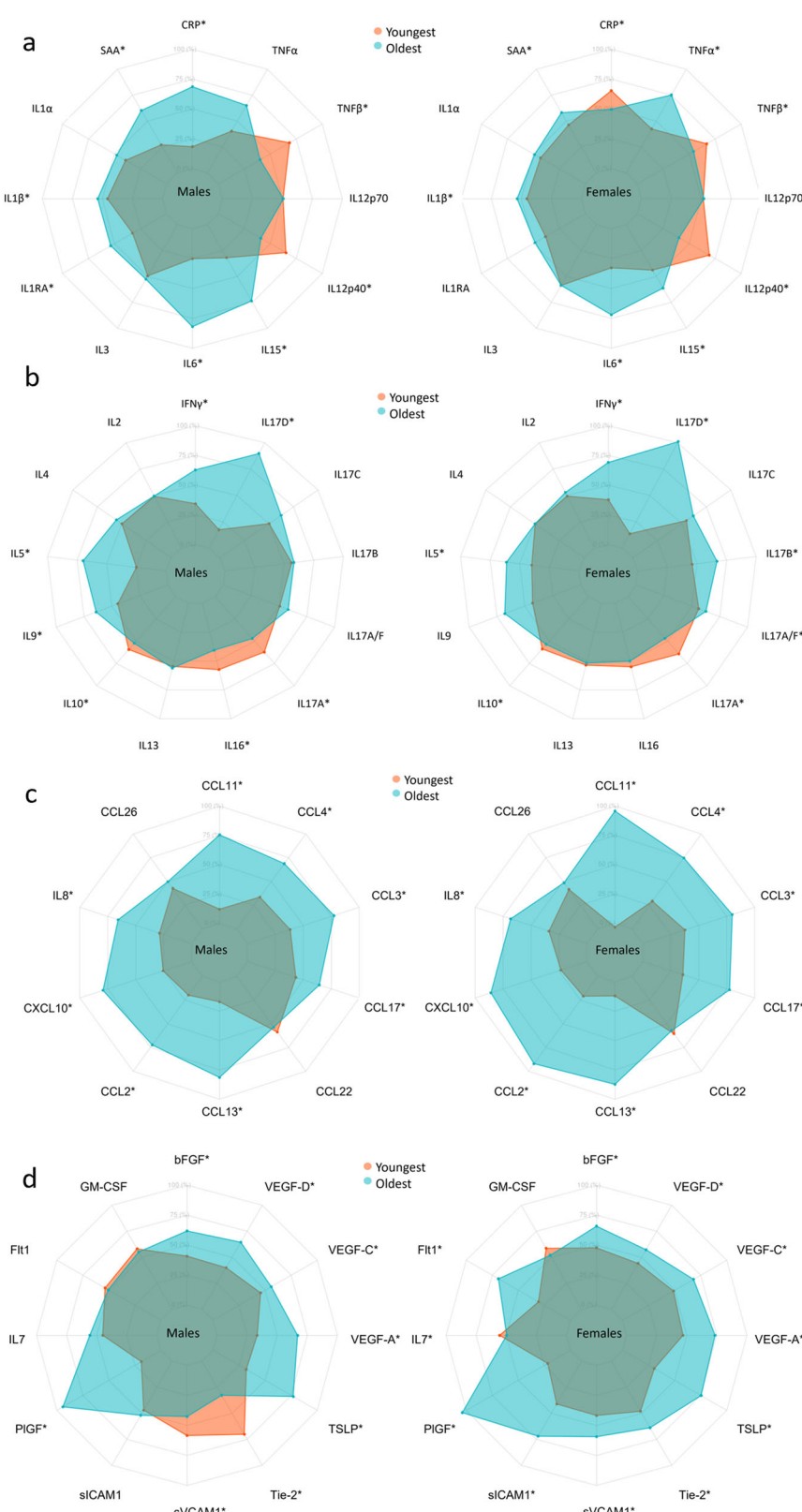

may impact the concentration of inflammatory markers[46]. Although participants provided information on previous smoking behavior, we did not assess its effect since we unfortunately have no knowledge on the date of cessation, and we chose not to include this data in the analysis.

For many of the investigated biomarkers, we observed sex-differences or interactions between sex and BMI but not between sex and smoking.

These findings strongly emphasize that sex, and especially the interaction between sex and age and the effects of menopause, must be considered when using biomarkers in tailored patient care or decision support tools. Some of the notable sex interactions with age and sex interactions with BMI were high concentrations of CRP in young female participants and high concentrations of Tie-2 and TNF-α in male participants. Our findings show that

**Fig. 6 | Radar charts of the median of the scaled concentration of biomarkers for BMI groups.** Axis percentage spans from the first to the third quartile within each biomarker. Normal BMI in blue, overweight BMI in magenta, obese BMI in red/orange. **a** is proinflammatory group, **b** is T cell-derived group, **c** is chemokine group, and **d** is growth factors and vascular group. * marks confidence intervals not spanning 0 for the difference between overweight and obese compared to normal, # marks the same for just one group.

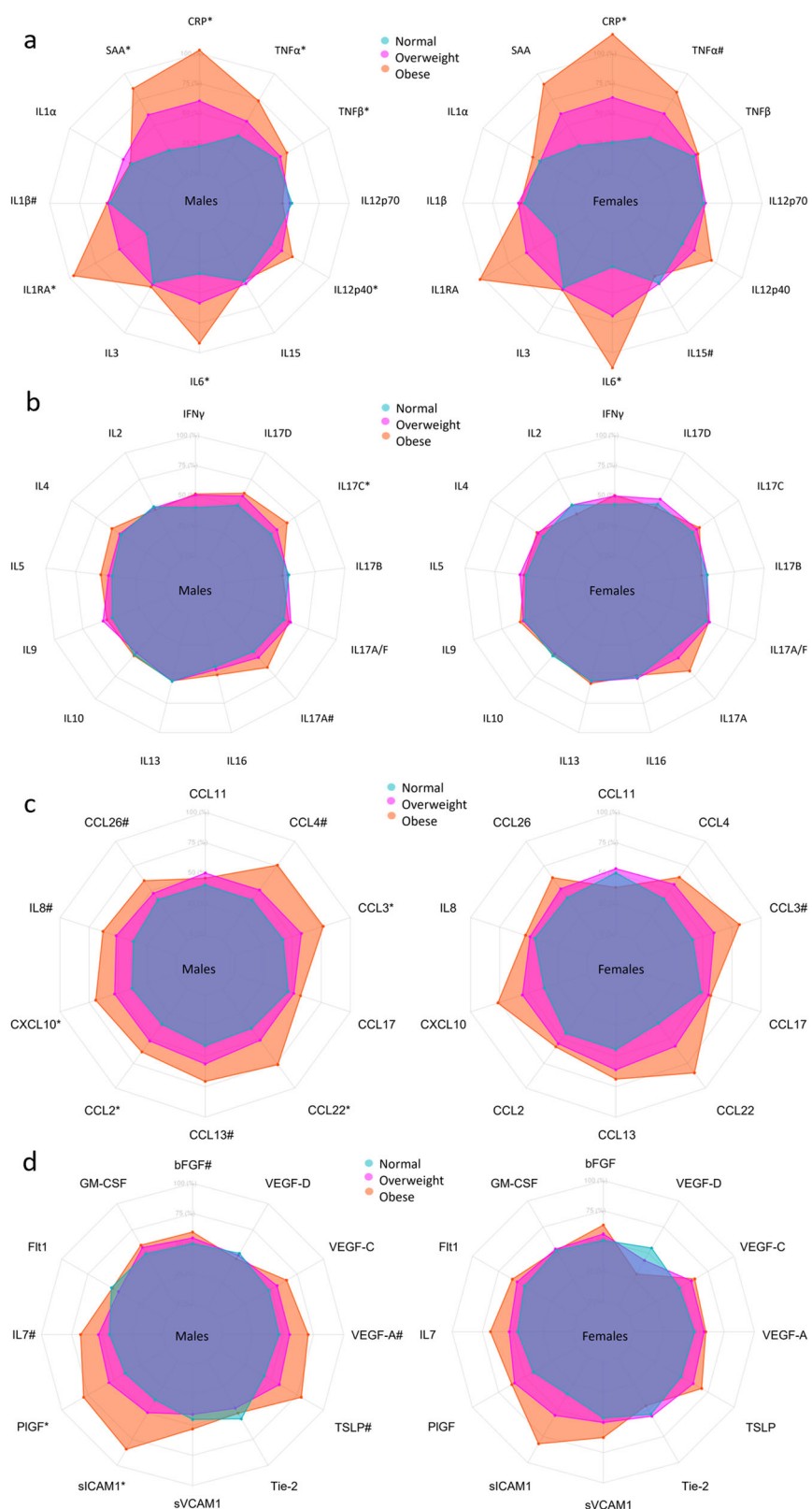

concentrations of several biomarkers changed differently with age in the two sexes. The observed age and sex-differences in biomarkers highlight the need for careful planning of study designs when utilizing biomarker measures such as nested case-control designs matching on age and sex. Although CCL13, CCL26, IL-1α, IL-5, IL-12p70, IL-17B, IFNγ, SAA, and CRP were changed statistically significantly with smoking in one sex only,

the interaction was not statistically significant after adjusting the significance level for mass testing, which naturally does not exclude the existence of a sex-specific effect of smoking. Sex-specific effects are likely as differences in stress response have been described[47], the X chromosome holds immune-related genes[48], immune cells have estrogen receptors[49], a sex difference in immune response has been shown in rats[50], and the

**Fig. 7 | Radar charts of the median of the scaled concentration of biomarkers for smokers vs non-smokers.** Axis percentage spans from the first to the third quartile within each biomarker. **a** is proin-flammatory group, **b** is T cell-derived group, **c** is chemokine group, and **d** is growth factors and vascular group. * marks a statistically significant difference between smokers and non-smokers.

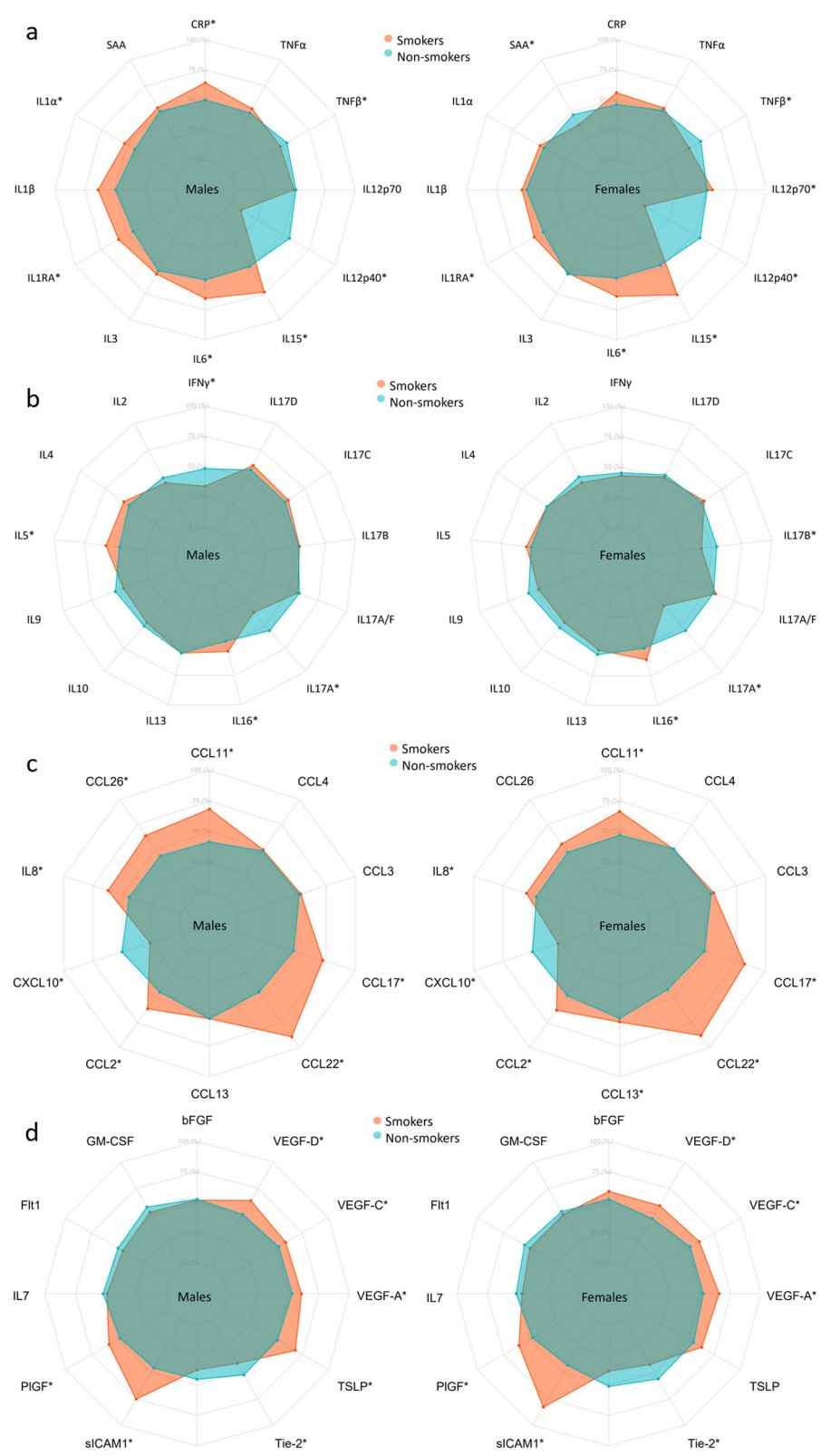

metabolism of nicotine has been found to be faster in females[51]. CCL26 is expressed by lung endothelial cells, CCL13 is induced by IL-1, and the known associations between CRP and estrogen speak for a sex-specific effect of smoking, likely related to sex-hormonal differences.

The link between circadian rhythm and inflammation is well known, but for most of the investigated biomarkers, much remains to be learned[19]. Our study revealed that older blood donors preferred donating blood slightly earlier in the day than young donors (Supplementary Fig. 4). The time of day may thus both have reflected the participant's age, influenced biomarker concentrations through the circadian rhythms, and reflected the time between sample draw and centrifugation and freezing, as turnaround times may vary slightly throughout the day. We

determined that adjusting for only age, rather than also time of day, was sufficient in our models.

This is a large study providing directly usable concentrations for several inflammatory and vascular stress biomarkers in healthy individuals. However, we do acknowledge some limitations to the study setup. Samples were exposed to a freeze-thaw cycle when being aliquoted. Because this was the same for all samples it should not bias our findings of biomarker associations. However, future studies that may potentially use the reported concentrations as a reference should keep this in mind. We do not provide reference intervals in a traditional sense but provide an expectable distribution of concentrations and the impact of normal demographic and lifestyle factors in healthy individuals. For several biomarkers, a large percentage of measurements were below the detection limit (Supplementary Data 1), which is a general challenge when assessing inflammation and vascular stress biomarkers among healthy individuals. This makes our data sensitive to how these values were handled, as discussed in the methods section. Given the participants' healthy state, the measurements below the lower limit of detection were partly expected[9]. Furthermore, when inspecting QQ-plots for the linear models, we did not observe a perfect Gaussian distribution of residuals for all biomarkers. Primarily, we observed data to be slightly leptokurtic for assays, with a large proportion of samples below the fit curve range. Also, we could not accurately account for the length of time from sample draw to freezing or for the fact that procedures may have changed slightly over the course of the study as we picked samples collected over an 11-year period. Although donation sites have applied the same framework, regional and site-specific differences in logistics may have influenced the time from sample draw to centrifugation or freezing. We found regional differences for 31 biomarkers (Supplementary Data 4), although for 16 biomarkers, the difference was limited to a single region, and the effects were generally small. These variations may relate to differences in sample handling, but there are also other differences between the regions, such as demographic and socioeconomic factors, that possibly contribute to the observed variations. Socioeconomic data was unavailable for this study. Samples from biomarker studies are often drawn from fasting participants. This is not possible with blood donors as they are advised to eat and drink before donation. Lifestyle factors are well-known to play an important role in the pathophysiology of numerous diseases, such as metabolic disease, cardiovascular disease, diabetes, lung diseases, and infections[52–55]. Our questionnaire data allowed for model adjustments and for investigation of the influence of common demographic and lifestyle factors, not available in health registers, on biomarker concentrations. The size of our study population provided sufficient power to stratify by these common variables. Possible conflicts between our observations and those reported in previous studies may also be explained by different compositions of study populations and power (n) and the healthy donor effect, i.e., only those who are healthy can become and remain donors. Though the healthy donor effect may limit the generalizability of the findings, the fact that the population of blood donors is healthier than the normal healthy population[56,57] reduces the risk of confounding diseases affecting the results and thus makes this study ideal as a reference population for studies requiring healthy reference values and for studying how demographic and lifestyle factors affect a given biomarker. Notably, the Danish blood donors do not perfectly represent the background population. They are generally healthier, have a higher income and education level than the background population, and the prevalence of donors is higher in urban areas. Additionally, a larger proportion has at least one Danish parent, and for males there is an association with cohabitation with a female[58]. The selection of healthy donors also causes the healthy donor effect to increase with age, as only the healthy remain donors. Also, even though we have been able to include possible covariates that we deemed to have a direct causal effect on biomarker concentrations, we cannot rule out the potential bias from unmeasured confounders.

The provided concentrations may help guide researchers and clinicians in, e.g., study setup, power calculations, assay designs, future disease detection, etc., including the need to adjust for sex, age, BMI, and smoking. As proteomics data become available, we plan to investigate how these data behave in our cohort and how they correlate with the biomarkers reported here. Studies using measurement methods with relative intensities or abundances may not benefit from the listed expected concentrations but can still benefit from the listed percentage changes and general trends described.

## Conclusion

In this study, we measured a wide range of biomarkers in a large cohort of 9876 sex- and age-balanced blood donors to reveal the impact of normal demographic and lifestyle factors on concentrations of inflammatory and vascular stress biomarkers in healthy individuals. We found that concentrations of several biomarkers increased with age, with some variation in the magnitude of increase between the sexes. Concentrations of proinflammatory biomarkers generally increased with BMI and smoking, with significant sex-differences observed. Conversely, multiple T cell-derived biomarker concentrations were lower or similar in smokers when compared to non-smokers. Together, our results emphasize that plasma levels of inflammatory and vascular stress biomarkers are highly associated with demographic and lifestyle factors, including sex, age, BMI, and smoking status. This emphasizes that an improved understanding of such associations in both healthy individuals and patients is instrumental to advance our understanding of disease development (related to, e.g., smoking and obesity) as well as facilitating the development of precision diagnostics and tailored care and decision support tools based on the investigated biomarkers.

## Data availability

The DBDS is a platform for studies carried out by the Danish blood centers and collaborators. The study is managed by a steering committee who respond to enquiries regarding collaboration. The blood donors participate in the DBDS to increase the scope of their donation, i.e. to help produce valuable research for the benefit of future patients. Additional information can be found on our home page [http://www.dbds.dk]. We invite researchers to collaborate by contacting the steering committee [info@dbds.dk]. Data access requires that projects and applicants obtain permission from the Regional Committees on Health Research Ethics and the Danish Data Protection Agency [http://www.datatilsynet.dk]. Source data for figures in the main manuscript have been uploaded and are available as Supplementary Data 8–11.

## Code availability

Data analysis code is available online[59].

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

## Acknowledgements

We take this opportunity to express our gratitude to the Danish blood donors for their valuable participation in the Danish Blood Donor Study and to the staff at the blood centers for making this study possible. The Danish Blood Donor Study is funded by the Danish Council for Independent Research—Medical Sciences (8020-00403B), the Danish Administrative Regions and Bio and Genome Bank Denmark, and the Danish Blood Donor Research Foundation. BK and KAK are supported by a grant from BERTHA—the Danish Big Data Centre for Environment and Health, which, in turn, is funded by the Novo Nordisk Foundation Challenge Program (grant NNF17OC0027864). ATL, DW, KB, KSB, TFH, and SB acknowledge the Novo Nordisk Foundation (grants NNF17OC0027594 and NNF14CC0001). MN has received funding from the European Union's Horizon 2020 research and innovation program under grant agreement No 101017562. HSN has received funding from the Augustinus Foundation (19-3829). MD received funding from the Danish Council for Independent Research (09-069412). The funders had no role in the design or execution of the study.

## Author contributions

A.J., B.L.M., B.K., C.B.P., C.E., C.E.S., E.S., H.B., H.S.N., H.H., H.U., K.B., K.S.B., K.N., L.V.K., M.D., M.N., M.T.B., N.G., N.Ø., O.B.P., P.Je, P.Jo, R.F.S., R.L.J., S.R.O., S.B., S.J., T.F.H., T.H. and T.L.S. advised on study design and participant selection. L.H., M.H.L. and M.S. coordinated sample identification. B.K., B.P., J.A.M., J.G., L.J.H., M.H.L., S.S.B. and T.K.L. supervised, coordinated, and performed biomarker measurements. B.K., J.G., M.H.L., R.L.J., J.D. and A.T.L. implemented quality control measures. B.K., R.L.J. and J.D. performed the statistical analysis and plotting. A.T.L., C.E., D.W., J.K.B., J.G., K.B., K.R., R.L.J., and S.R.O. counseled on statistics and data presentation. B.K., C.E., E.S., H.U., H.H., J.D., J.G., K.M.D., K.R.N., L.H., K.A.K., L.W.T., M.H.L., M.S., M.T.B., O.B.P., S.M., and S.R.O. supervised or assisted in inclusions and data flow in the DBDS. B.K. prepared the initial draft of the manuscript. All authors critically read and revised the manuscript.

## Competing interests

The authors declare no competing interests.

## Additional information

Bertram Kjerulff ®[1,2,3] ✉, Joseph Dowsett ®[4], Rikke Louise Jacobsen ®[4], Josephine Gladov ®[1,2,3], Margit Hørup Larsen[4], Agnete Troen Lundgaard ®[5], Karina Banasik ®[5], David Westergaard ®[5,6], Susan Mikkelsen[1], Khoa Manh Dinh ®[1], Lotte Hindhede[1], Kathrine Agergård Kaspersen ®[1,3], Michael Schwinn ®[4], Anders Juul ®[6,7,8], Betina Poulsen[4], Birgitte Lindegaard[6,9], Carsten Bøcker Pedersen ®[3,10], Clive Eric Sabel ®[3,11,12], Henning Bundgaard[6,13], Henriette Svarre Nielsen ®[6,14,15], Janne Amstrup Møller[4], Jens Kjærgaard Boldsen[1,3], Kristoffer Sølvsten Burgdorf[5], Lars Vedel Kessing[6,16], Linda Jenny Handgaard[4], Lise Wegner Thørner[4], Maria Didriksen ®[4], Mette Nyegaard ®[17], Niels Grarup ®[18], Niels Ødum[19], Pär I. Johansson ®[4,15], Poul Jennum[6,20], Ruth Frikke-Schmidt ®[6,21], Sanne Schou Berger ®[22], Søren Brunak ®[5], Søren Jacobsen ®[6,23], Thomas Folkmann Hansen ®[5,24], Tine Kirkeskov Lundquist[4], Torben Hansen ®[18], Torben Lykke Sørensen[6,25], Torben Sigsgaard ®[3,11], Kaspar René Nielsen[26], Mie Topholm Bruun ®[27], Henrik Hjalgrim ®[6,28,29,30], Henrik Ullum[31], Klaus Rostgaard ®[28,29], Erik Sørensen[4], Ole Birger Pedersen ®[6,32,33], Sisse Rye Ostrowski ®[4,6,33] & Christian Erikstrup ®[1,2,3,33]

¹Department of Clinical Immunology, Aarhus University Hospital, Aarhus, Denmark. ²Department of Clinical Medicine, Aarhus University, Aarhus, Denmark. ³BERTHA Big Data Centre for Environment and Health, Aarhus University, Aarhus, Denmark. ⁴Department of Clinical Immunology, Copenhagen University Hospital, Rigshospitalet, Copenhagen, Denmark. ⁵Translational Disease Systems Biology, Novo Nordisk Foundation Center for Protein Research, Faculty of Health and Medical

Sciences, University of Copenhagen, Copenhagen, Denmark. [6]Department of Clinical Medicine, Faculty of Health and Medical Sciences, University of Copenhagen, Copenhagen, Denmark. [7]Department of Growth and Reproduction, Copenhagen University Hospital — Rigshospitalet, Copenhagen, Denmark. [8]International Center for Research and Research Training in Endocrine Disruption of Male Reproduction and Child Health (EDMaRC), Rigshospitalet, University of Copenhagen, Copenhagen, Denmark. [9]Department of Pulmonary and Infectious Diseases, Copenhagen University Hospital—North Zealand, Hillerød, Denmark. [10]National Centre for Register-based Research, Aarhus BSS, Aarhus University, Aarhus, Denmark. [11]Department of Public Health, Aarhus University, DK-8000 Aarhus, Denmark. [12]School of Geography, Earth and Environmental Sciences, University of Plymouth, Plymouth PL4 8AA, UK. [13]The Heart Center, Rigshospitalet, Copenhagen University Hospital, Copenhagen, Denmark. [14]Recurrent Pregnancy Loss Unit, Capital Region, Copenhagen University Hospitals, Hvidovre and Rigshospitalet, Copenhagen, Denmark. [15]Department of Obstetrics and Gynecology, Copenhagen University Hospital, Hvidovre, Denmark. [16]Copenhagen Affective Disorder Research Center (CADIC), Psychiatric Center Copenhagen, Copenhagen, Denmark. [17]Department of Health Science and Technology, Aalborg University, Aalborg, Denmark. [18]Novo Nordisk Foundation Center for Basic Metabolic Research, Faculty of Health and Medical Sciences, University of Copenhagen, Copenhagen, Denmark. [19]LEO Foundation Skin Immunology Research Center, Department of Immunology and Microbiology, University of Copenhagen, Copenhagen, Denmark. [20]Danish Center for Sleep Medicine, Department of Clinical Neurophysiology, Rigshospitalet, Copenhagen, Denmark. [21]Department of Clinical Biochemistry, Copenhagen University Hospital—Rigshospitalet, Copenhagen, Denmark. [22]Centre for Diagnostics, DTU Health Technology, Technical University of Denmark, 2800 Kgs. Lyngby, Denmark. [23]Copenhagen Lupus and Vasculitis Clinic, Center for Rheumatology and Spine Diseases, Rigshospitalet, Copenhagen, Denmark. [24]Danish Headache Center and Danish Multiple Sclerosis Center, Copenhagen University Hospital, Rigshospitalet Glostrup, Glostrup, Denmark. [25]Clinical Eye Research Division, Department of Ophthalmology, Zealand University, Hospital, Roskilde, Denmark. [26]Department of Clinical Immunology, Aalborg University Hospital, Aalborg, Denmark. [27]Department of Clinical Immunology, Odense University Hospital, Odense, Denmark. [28]Department of Epidemiology Research, Statens Serum Institut, Copenhagen, Denmark. [29]Danish Cancer Society Research Center, Danish Cancer Society, Copenhagen, Denmark. [30]Department of Hematology, Copenhagen University Hospital, Copenhagen, Denmark. [31]Statens Serum Institut, Copenhagen, Denmark. [32]Department of Clinical Immunology, Zealand University Hospital, Køge, Denmark. [33]These authors contributed equally: Ole Birger Pedersen, Sisse Rye Ostrowski, Christian Erikstrup. ✉e-mail: berkje@rm.dk

