## [Peer Review File · Communications Medicine]

Reviewers' comments:

Reviewer #1 (Remarks to the Author):

Kjerfulff et al.'s manuscript "Sex, age, BMI, and smoking associations with 47 inflammatory and vascular stress biomarkers in 9,876 blood donors" assesses how demographic and common lifestyle factors are associated with selected circulating inflammatory and vascular stress biomarkers in a large Danish cohort of healthy adults. Kjerfulff et al. show a general increase in concentrations of inflammation and vascular stress biomarkers with higher age, among people with higher BMI compared to lower BMI, and among smokers compared with non-smokers. Moreover, they show that biological sex interacts these factors for many of these circulating biomarkers. The authors note that in particular, the observed age and sex differences in biomarkers could have potential implications for designing studies with biomarker measurements, such as nested case-control designs, that match on age and sex.

Overall, the motivation of this study to better characterize the profiles of circulating inflammatory and vascular stress biomarkers among healthy adults based on several demographic/lifestyle factors was well-justified, and many analyses were performed to try to address this question. My comments and questions primarily pertain to decisions made in the linear regression modeling, and the measurement of the biomarkers.

1) For their primary linear models (assessing the association between sex and biomarker concentrations, BMI and biomarker concentrations, and smoking status and biomarker concentrations), the authors adjust for a number of different variables including: age, sex (for the models in which sex was not an exposure), smoking (for the models in which smoking was not an exposure), BMI (for the models in which BMI was not an exposure), sampling time of day, region, sample storage time, and measurement date. While some of these variables in some models represent "traditional" confounders (i.e., predictors of both the exposure and outcome), many of these variables only seem to be predictors of the outcome, but not the exposure (i.e., sample time of day, measurement date, sample storage time, in some models, region and age). As described in PMID: 19525685 (see page 7), these are examples of "unnecessary adjustments", and adjusting for predictors of the outcome will not bias the effect estimate, but can improve the precision of the association. If improving the precision of the association is the motivation for including these variables in the models, I think the authors should explicitly state this and the causal structure of their models, so it is clear to readers that they do not interpret these variables as traditional confounders. Additionally, in the model in which sex is the exposure, BMI and smoking are not plausible predictors of sex (the exposure), but sex (the exposure) could be predictors of these variables. Similarly, in the model in which smoking status is the exposure, it is possible that BMI could be a predictor of smoking status (e.g. if someone is smoking as an attempt to lose weight). But it could also be a mediator of the smoking-biomarker relationship. In both of these cases, it is plausible that these mediators (BMI, smoking) could be associated with one of the other variables associated with the outcome (e.g. age), or another unmeasured confounder. This is an example of mediator-outcome confounding, and could result in a biased estimate if not all mediator-outcome confounders (e.g. unmeasured confounders) are accounted for (see PMID: 24019424, page 1513). Overall, the authors should articulate more clearly their reasonings for including the given covariates in their models, and if they choose to keep in these possible mediators, they should describe the causal structure of their models and mention this potential source of bias. Finally, on lines 444-445, the authors state: "Our study revealed that older blood donors preferred donating blood slightly earlier in the day than young donors, emphasizing that both age and time of day were important

factors to adjust for in our models.” If, as the authors suggest, age is a predictor of time of blood draw, the minimally sufficient adjustment set for covariates only needs to include age, not age and sample time of day. Therefore, I disagree with this statement. I have included a document with DAGs for each of the three models to illustrate the apparent causal structure of these variables, based on how the linear modeling was described in the manuscript text.

2) On lines 149-150, the authors write: “Biomarker measurements were performed using the Meso Scale Discovery (MSD) V-PLEX Human Biomarker 54-plex kit.” Based on my internet browsing, I surmised that the MSD is an electroluminescence-based platform (https://www.mesoscale.com/en/technical_resources/our_technology/ecl/), but I think it would be useful for readers, especially those who may be more familiar with mass spectrometry-based approaches for quantifying targeted and untargeted biomarkers, to directly state the method by which these inflammatory and vascular stress markers are measured in the MSD assay.

3) Relatedly, the authors note that one of the novel aspects their study is the presentation of absolute biomarker concentrations in specific age and sex-stratified subgroups in their relatively large sample of healthy individuals. The fact that they can report absolute biomarker concentrations seems to be a result of using this particular platform/assay. It would be informative if the authors could comment on how these results should be interpreted in the context of large scale proteomics and metabolomics studies that report relative abundance or intensities because this is what proteomics/metabolomics platforms provide. For example, on lines 486-487, the authors state “...thus makes this study ideal as a reference population for studies requiring healthy reference values and for studying how demographic and lifestyle factors affect a given biomarker.” However, if future studies use a platform/assay that doesn’t provide absolute values, the “healthy reference values” provided here might not apply; should the general trends observed here be used as the “reference” instead?

4) In their discussion of potential limitations, the authors should also consider mentioning the fact that the population represented by this cohort was likely relatively homogeneous with respect to other sociodemographic factors such as race/ethnicity and socioeconomic status. Therefore, in order to determine whether these “healthy reference values” are truly representative “references,” similar studies should be performed in other diverse populations.

Reviewer #2 (Remarks to the Author):

Kjerulff et al provide a comprehensive and detailed description of the association between common phenotypes (e.g., sex, age, BMI and smoking) and 47 plasma biomarkers, most of them being cytokines, in a large sample of blood donor in Denmark. Even if the overall conclusion has little novelty, as pointed by the authors (age, male sex, BMI, and smoking were associated with higher concentration of pro-inflammatory biomarkers and lower concentration of anti-inflammatory biomarkers), the study has its merits. The authors make a nice and valid argumentation, in my opinion, for the need of information on the relation between biomarkers and phenotypes as we strive for precision medicine.

Major comment:

1) In the abstract, the authors wrote “Using regression model, we examined the association between biomarkers and sex, age, Body Mass Index (BMI), smoking, and time-of-day variation”. However, it is not described in the Results section about the regression of biomarkers on time-of-day variation. Time-of-day variations was rather used a technical covariate for adjustment. I suggest that the authors either remove the wording “time-of-day variation” from the abstract or present some of the linear regression results in the Results section (not enough with plots of smooth conditional means in the Supplements).

Minor comments,

1) Was any negative control added to the rounds of biomarker measurements?

2) Was there information available on previous smokers? If so, would it be worth adding them as a separate category from non-smokers?

3) Results section, page 10, line 310, “The effect of obesity on concentrations of CCL11 and CCL4 differed by sex (see interaction terms in Figure 3 and the Appendix).” Could the authors be more specific? What was the difference between sexes?

4) Results section, page 10, lines 315-316, “The effect of BMI on concentrations of VEGF-D differed between males and females although no association was found with BMI for VEGF-D, as displayed by the interaction terms in Appendix”. I don’t understand this. The interaction term for ObeseXsex in the Appendix is quite low and in bold ($P = 4.07 \times 10^{-6}$). Perhaps inverting the sentence to: “Although no association was found with BMI for VEGF-D, the effect of BMI on concentrations of VEGF-D differed between males and females, as displayed by the interaction terms”?

5) I strongly encourage the authors to publicly share the R codes used for the statistical analyses.

Reviewer #3 (Remarks to the Author):

Kjerulff et al. present an analysis of the Danish Blood Donor Study (DBDS) biobank data to investigate associations between demography and lifestyle factors with 47 inflammatory and vascular stress biomarkers in 9,876 individuals. This is an interesting, largely descriptive study which offers an interesting overview on how these biomarkers vary with specific variables collected in epidemiological settings. Below are some comments which the authors should address to improve reporting of their results:

1. Consistency is pivotal to make this analysis easier to follow for readers. On that sense, it is unclear why the interaction between smoking and sex was not reported in the analysis. Whilst the sexual dimorphisms for these biomarkers are apparent, their biological relevance should be further discussed, with evidence from previous studies supporting these associations.

2. The statistical analysis section is very hard to follow. Please consider organizing the analytical frameworks with subheadings and providing a diagram which shows the methods used for data processing and modeling in the study. This would make the results section also more articulate.

3. Authors show regional differences in the distribution of biomarkers and attribute these differences to unmeasured confounders. Given that this appears to be a large source of heterogeneity, mixed effects linear models could be fitted using random intercepts for each region,

to account for its influence in modifying the magnitude of these effects.

4. Since a strength of this analysis is its large sample size of healthy individuals, it is important to further highlight the stark differences presented in radar plots in Supplementary Materials. Authors should consider shifting these figures to the main manuscript and discussing the relevance of these differences. Some can be attributable to aging, the influence of adiposity and smoking and its pathophysiological relevance should also be highlighted in the manuscript. This is currently lacking.

5. The authors mention residual diagnostics in the discussion section, but these are not described in the methods section. Please provide a detailed section on how model diagnostics were conducted for all fitted linear models.

6. Information on missing data is very detailed but using as a reference the details provided in the Methods section, it is unclear whether multiple imputation was also performed for smoking status and BMI. Was a complete case analysis conducted? If so, authors are suggested to consider multiple imputation and pooled modeling combining model estimates with Rubin's rules.

7. Were interactions between BMI and age, and between smoking status and age assessed? For some biomarkers, differences in age are quite apparent and it would be interesting to explore whether the associations with BMI and/or smoking are also modified by age.

Responses to Referees

Referee expertise:

Referee #1: Biomarkers, biostatistics

Referee #2: Biomarkers, cardiometabolic disease, Biostatistics

Referee #3: Cardiovascular/metabolic health, clinical, epidemiology, biomarkers

Dear Editor and Reviewers,

We would like to express our sincere gratitude for the thoughtful and constructive feedback you have provided on our manuscript. Your insights have been invaluable in enhancing both the quality and conciseness of our work. We have carefully considered each comment and have made revisions accordingly. Below, you will find a detailed response to each point raised, outlining the changes we have implemented to address your concerns. Note that the line numbers given below for changes made in the manuscript refer to the clean-version of the manuscript without track-changes.

Reviewers' comments:

Reviewer #1 (Remarks to the Author):

Kjerfulff et al.'s manuscript "Sex, age, BMI, and smoking associations with 47 inflammatory and vascular stress biomarkers in 9,876 blood donors" assesses how demographic and common lifestyle factors are associated with selected circulating inflammatory and vascular stress biomarkers in a large Danish cohort of healthy adults. Kjerfulff et al. show a general increase in concentrations of inflammation and vascular stress biomarkers with higher age, among people with higher BMI compared to lower BMI, and among smokers compared with non-smokers. Moreover, they show that biological sex interacts these factors for many of these circulating biomarkers. The authors note that in particular, the observed age and sex differences in biomarkers could have potential implications for designing studies with biomarker measurements, such as nested case-control designs, that match on age and sex.

Overall, the motivation of this study to better characterize the profiles of circulating inflammatory and vascular stress biomarkers among healthy adults based on several demographic/lifestyle factors was well-justified, and many analyses were performed to try to address this question. My comments and questions primarily pertain to decisions made in the linear regression modeling, and the measurement of the biomarkers.

1) For their primary linear models (assessing the association between sex and biomarker concentrations, BMI and biomarker concentrations, and smoking status and biomarker concentrations), the authors adjust for a number of different variables including: age, sex (for the models in which sex was not an exposure), smoking (for the models in which smoking was not an exposure), BMI (for the models in which BMI was not an exposure), sampling time of day, region, sample storage time, and measurement date. While some of these variables in some models represent "traditional" confounders (i.e., predictors of both the exposure and outcome), many of these variables only seem to be predictors of the outcome, but not the exposure (i.e., sample time of day, measurement date, sample storage time, in some models, region and age). As described in PMID: 19525685 (see page 7), **these are examples of "unnecessary adjustments", and adjusting for predictors of the outcome will not bias the effect estimate, but can improve the precision of the association. If improving the precision of the association is the motivation for including these**

variables in the models, I think the authors should explicitly state this and the causal structure of their models, so it is clear to readers that they do not interpret these variables as traditional confounders. Additionally, in the model in which sex is the exposure, BMI and smoking are not plausible predictors of sex (the exposure), but sex (the exposure) could be predictors of these variables. Similarly, in the model in which smoking status is the exposure, it is possible that BMI could be a predictor of smoking status (e.g. if someone is smoking as an attempt to lose weight). But it could also be a mediator of the smoking-biomarker relationship. In both of these cases, it is plausible that these mediators (BMI, smoking) could be associated with one of the other variables associated with the outcome (e.g. age), or another unmeasured confounder. This is an example of mediator-outcome confounding, and could result in a biased estimate if not all mediator-outcome confounders (e.g. unmeasured confounders) are accounted for (see PMID: 24019424, page 1513). **Overall, the authors should articulate more clearly their reasonings for including the given covariates in their models, and if they choose to keep in these possible mediators, they should describe the causal structure of their models and mention this potential source of bias.** Finally, on lines 444-445, the authors state: "Our study revealed that older blood donors preferred donating blood slightly earlier in the day than young donors, emphasizing that both age and time of day were important factors to adjust for in our models." **If, as the authors suggest, age is a predictor of time of blood draw, the minimally sufficient adjustment set for covariates only needs to include age, not age and sample time of day. Therefore, I disagree with this statement.** I have included a document with DAGs for each of the three models to illustrate the apparent causal structure of these variables, based on how the linear modeling was described in the manuscript text.

Thank you for these comments and the attached document with DAGs. We have elaborated on our motivation for including covariates in the *Statistics section* (line 178–185) and added a Figure S7 in the Supplementary material with DAGs for each of the three models to illustrate our view of the causal structure of the variables in the models.

Our intention was to obtain the most precise estimates of the effect of the exposure on the outcome. Therefore, in our linear models, we adjusted for possible covariates that we deemed to have a direct causal effect on the outcome, as illustrated in Figure S7 (page 29 in Supplementary).

In DAG (1), where sex is the exposure, we agree that BMI and smoking are not plausible predictors of sex. However, we consider them to be mediators that should be adjusted for. I.e., we consider the direct effect of sex to be the effect of primary interest in most situations, rather than the total or overall effect that would be estimated when leaving BMI and smoking out of the model.

In DAG (2), where BMI is the exposure, we are in full agreement with the reviewer's suggested DAG.

DAG (3) with smoking as exposure, we are in full agreement with the reviewer's suggested DAG.

After consideration, we agree that it is unnecessary to also adjust for sample time-of-day when adjusting for age. Therefore, we have removed the covariate sample time-of-day from our models, as illustrated in Figure S7. We have made changes accordingly in the manuscript (lines 181–183, line 464), the abstract (line 58), figure legends of figures 2–4, and removed the column 'Circadian Rhythm' from the *Results Appendix with Details for each biomarker*.

Also, we have added a phrase regarding potential source of bias from unmeasured confounders as suggested (lines 506–508). All analyses and associated tables have been adjusted accordingly.

2) On lines 149-150, the authors write: "Biomarker measurements were performed using the Meso Scale Discovery (MSD) V-PLEX Human Biomarker 54-plex kit." Based on my internet browsing, I surmised that the MSD is an electroluminescence-based platform (https://www.mesoscale.com/en/technical_resources/our_technology/ecl/), but I think it would be useful for readers, especially those who may be more familiar with mass spectrometry-based approaches for quantifying targeted and untargeted biomarkers, to directly state the method by which these inflammatory and vascular stress markers are measured in the MSD assay.

Thank you for this comment, we agree that this information would make the paper more informative and easily understandable for readers not familiar with MSD. We now state that MSD is based on electrochemiluminescence (line 141).

3) Relatedly, the authors note that one of the novel aspects their study is the presentation of absolute biomarker concentrations in specific age and sex-stratified subgroups in their relatively large sample of healthy individuals. The fact that they can report absolute biomarker concentrations seems to be a result of using this particular platform/assay. It would be informative if the authors could comment on how these results should be interpreted in the context of large scale proteomics and metabolomics studies that report relative abundance or intensities because this is what proteomics/metabolomics platforms provide. For example, on lines 486-487, the authors state "...thus makes this study ideal as a reference population for studies requiring healthy reference values and for studying how demographic and lifestyle factors affect a given biomarker." However, if future studies use a platform/assay that doesn't provide absolute values, the "healthy reference values" provided here might not apply; should the general trends observed here be used as the "reference" instead?

This is an important point that merits discussion. We believe that studies lacking absolute concentrations can still gain insights from the general trends and percentage changes detailed in the extensive appendix. Accordingly, we have addressed this in the *Discussion* (line 512-514). However, it is worth noting that for the implementation of new biomarkers in screenings, ELISA-based assays are generally preferred over mass-spec based assays.

4) In their discussion of potential limitations, the authors should also consider mentioning the fact that the population represented by this cohort was likely relatively homogeneous with respect to other sociodemographic factors such as race/ethnicity and socioeconomic status. Therefore, in order to determine whether these "healthy reference values" are truly representative "references," similar studies should be performed in other diverse populations.

It is very true that the blood donors do not reflect the background population in every aspect. Education level and ethnicity are rightly two aspects in which they differ. We have expanded on this in the discussion and added a relevant reference on the differences between blood donors and the general Danish population (lines 501-506).

Reviewer #2 (Remarks to the Author):

Kjerulff et al provide a comprehensive and detailed description of the association between common phenotypes (e.g., sex, age, BMI and smoking) and 47 plasma biomarkers, most of them being cytokines, in a large sample of blood donor in Denmark. Even if the overall conclusion has little novelty, as pointed by the authors (age, male sex, BMI, and smoking were associated with higher

concentration of pro-inflammatory biomarkers and lower concentration of anti-inflammatory biomarkers), the study has its merits. The authors make a nice and valid argumentation, in my opinion, for the need of information on the relation between biomarkers and phenotypes as we strive for precision medicine.

Major comment:

1) In the abstract, the authors wrote "Using regression model, we examined the association between biomarkers and sex, age, Body Mass Index (BMI), smoking, and time-of-day variation". However, it is not described in the Results section about the regression of biomarkers on time-of-day variation. Time-of-day variations was rather used a technical covariate for adjustment. I suggest that the authors either remove the wording "time-of-day variation" from the abstract or present some of the linear regression results in the Results section (not enough with plots of smooth conditional means in the Supplements).

As detailed in the response to reviewer #1, we agree that it is unnecessary to include the covariate sample time-of-day in our models. Therefore, analyses have been repeated with the time-of-day variable omitted.

Minor comments,

1) Was any negative control added to the rounds of biomarker measurements?

The standard row (#8) included on each plate contained only diluent and thus functioned as a negative control. We have clarified that a standard containing diluent only served as a negative control (line 156–157).

2) Was there information available on previous smokers? If so, would it be worth adding them as a separate category from non-smokers?

We have the information on former smoking but lack details on when smoking ceased. Given that timespan from smoking cessation to biomarker measurement has impact on inflammatory marker concentrations (PMID: 24863424 and 28164227), and because we do not have this specific information, we chose not to separate these groups. The interpretation of such results would be fraught with uncertainty and would require an additional table for each biomarker. We have added a sentence to draw attention to this interesting perspective (line 437-439).

3) Results section, page 10, line 310, "The effect of obesity on concentrations of CCL11 and CCL4 differed by sex (see interaction terms in Figure 3 and the Appendix)." Could the authors be more specific? What was the difference between sexes?

Thank you for pointing this out, we have elaborated in the text (line 304). The effect of obesity was greater in males for CCL4 (interaction between obese and sex 1.091[1.029–1.156], p=0.0035, appendix page 64) and CCL11 (interaction between obese and sex 1.077[1.028–1.129], p=0.0019, Appendix page 65). Only the p-value is given in the tables in Appendix and the estimates and 95-CI for the interactions are not reported for any markers.

4) Results section, page 10, lines 315-316, "The effect of BMI on concentrations of VEGF-D differed between males and females although no association was found with BMI for VEGF-D, as displayed by the interaction terms in Appendix". I don't understand this. The interaction term for ObeseXsex in the

Appendix is quite low and in bold ($P = 4.07 \times 10^{-6}$). Perhaps inverting the sentence to: “Although no association was found with BMI for VEGF-D, the effect of BMI on concentrations of VEGF-D differed between males and females, as displayed by the interaction terms”?

Thank you for noting this, the suggested wording is clearer and has been inserted in the text (lines 309–310).

5) I strongly encourage the authors to publicly share the R codes used for the statistical analyses. We have made the code available using Zenodo and added a sentence with DOI in the relevant section of Data Availability (lines 555–556).

Reviewer #3 (Remarks to the Author):

Kjerulff et al. present an analysis of the Danish Blood Donor Study (DBDS) biobank data to investigate associations between demography and lifestyle factors with 47 inflammatory and vascular stress biomarkers in 9,876 individuals. This is an interesting, largely descriptive study which offers an interesting overview on how these biomarkers vary with specific variables collected in epidemiological settings. Below are some comments which the authors should address to improve reporting of their results:

1. Consistency is pivotal to make this analysis easier to follow for readers. On that sense, it is unclear why the interaction between smoking and sex was not reported in the analysis. Whilst the sexual dimorphisms for these biomarkers are apparent, their biological relevance should be further discussed, with evidence from previous studies supporting these associations.

Thank you for pointing this out. We have inserted the information regarding interaction between smoking and sex in the text under *Results* (line 350–353). Also, we have included a paragraph discussing the biologic relevance as suggested (lines 450–457).

2. The statistical analysis section is very hard to follow. Please consider organizing the analytical frameworks with subheadings and providing a diagram which shows the methods used for data processing and modeling in the study. This would make the results section also more articulate.

Thank you for the suggestion. We have reorganized the *Statistics section* with subheadings, which we believe enhances the clarity and readability of the *Methods* text (lines 178–209).

3. Authors show regional differences in the distribution of biomarkers and attribute these differences to unmeasured confounders. Given that this appears to be a large source of heterogeneity, mixed effects linear models could be fitted using random intercepts for each region, to account for its influence in modifying the magnitude of these effects.

We already account for regional differences through the use of a fixed effects model. Transitioning to a random effects model may offer only a marginal difference, if any, and we assert that the fixed effects model more precisely mirrors the full variation attributed to regional differences. For more details on our choice of adjustments, please also see our response to reviewer #1, where we describe the included adjustments illustrated with DAGs.

4. Since a strength of this analysis is its large sample size of healthy individuals, it is important to further highlight the stark differences presented in radar plots in Supplementary Materials. Authors should consider shifting these figures to the main manuscript and discussing the relevance of these differences. Some can be attributable to aging, the influence of adiposity and smoking and its pathophysiological relevance should also be highlighted in the manuscript. This is currently lacking.

Thank you. We have moved the radar plots from the *Supplementary* to the main text as suggested, now as Figures 5–7. Some of the differences may be influenced by ageing; however, as the adiposity and smoking models from which the significance of the differences have been drawn were adjusted for age, this should be somewhat alleviated. To highlight the pathophysiological relevance, we have added lines 490–491.

5. The authors mention residual diagnostics in the discussion section, but these are not described in the methods section. Please provide a detailed section on how model diagnostics were conducted for all fitted linear models.

Thank you for this point. We have now inserted detailed information on how model diagnostics were conducted for all fitted linear models (lines 186–187).

6. Information on missing data is very detailed but using as a reference the details provided in the Methods section, it is unclear whether multiple imputation was also performed for smoking status and BMI. Was a complete case analysis conducted? If so, authors are suggested to consider multiple imputation and pooled modeling combining model estimates with Rubin's rules.

We agree this is unclear. However, as explained above we have decided to omit sample time-of-day from our analysis model. The updated results and tables now reflect analyses based on case-complete participants. We have updated the manuscript accordingly in lines 179–180.

7. Were interactions between BMI and age, and between smoking status and age assessed? For some biomarkers, differences in age are quite apparent and it would be interesting to explore whether the associations with BMI and/or smoking are also modified by age.

In this study, we focus on exploring the direct associations between demographic and lifestyle factors and concentrations of inflammatory and vascular stress biomarkers. A priori, we chose to present age-stratified results, thereby foregoing additional interaction analyses involving age and BMI, or age and smoking. Nonetheless, we enriched our analyses by including interactions between sex and BMI, as well as sex and smoking. The Supplementary Appendix provides detailed age-stratified data for each biomarker. Adding further interaction analyses could complicate the manuscript for a general readership by expanding our already comprehensive supplementary material. If the reviewer wishes for additional analyses on this aspect, we are open to conducting them.

REVIEWERS' COMMENTS:

Reviewer #1 (Remarks to the Author):

Kjerulff et al's manuscript "Sex, age, BMI, and smoking associations with 47 inflammatory and vascular stress biomarkers in 9,876 blood donors" evaluates how demographic and select lifestyle characteristics (smoking) are associated with a selected number of circulating inflammatory and vascular stress biomarkers in a cohort of healthy Danish adults. They demonstrate a general increase in inflammation and vascular stress biomarkers with age, smoking, and BMI, with some variation in magnitude of increase between sexes. Kjerulff et al. have responded to reviewers' comments, updating some analyses as necessary (e.g., removing time-of-day adjustment) and providing more details in the methods, results, and discussion as appropriate. I think these changes have made this a stronger manuscript and have no further comments or suggested analyses.

Reviewer #2 (Remarks to the Author):

I appreciate the revision made the authors and I have no further comments.